# CORA: A Deep Active Learning Covid-19 Relevancy Algorithm to Identify Core Scientific Articles

**Zubair Afzal**
Elsevier, The Netherlands
m.afzal.1@elsevier.com

**Vikrant Yadav**
Elsevier, The Netherlands
v.yadav@elsevier.com

**Olga Fedorova**
Elsevier, The Netherlands
o.fedorova@elsevier.com

**Vaishnavi Kandala**
Elsevier, The Netherlands
v.kandala@elsevier.com

**Janneke van de Loo**
Elsevier, The Netherlands
j.loo.1@elsevier.com

**Saber A. Akhondi**
Elsevier, The Netherlands
s.akhondi@elsevier.com

**Pascal Coupet**
Elsevier, The Netherlands
p.coupet@elsevier.com

**George Tsatsaronis**
Elsevier, The Netherlands
g.tsatsaronis@elsevier.com

## Abstract

Ever since the *COVID-19* pandemic broke out, the academic and scientific research community, as well as industry and governments around the world have joined forces in an unprecedented manner to fight the threat. Clinicians, biologists, chemists, bioinformaticians, nurses, data scientists, and all of the affiliated relevant disciplines have been mobilized to help discover efficient treatments for the infected population, as well as a vaccine solution to prevent further the virus' spread. In this combat against the virus responsible for the pandemic, key for any advancements is the timely, accurate, peer-reviewed, and efficient communication of any novel research findings. In this paper we present a novel framework to address the information need of filtering efficiently the scientific bibliography for relevant literature around *COVID-19*. The contributions of the paper are summarized in the following: we define and describe the information need that encompasses the major requirements for *COVID-19* articles' relevancy, we present and release an expert-curated benchmark set for the task, and we analyze the performance of several state-of-the-art machine learning classifiers that may distinguish the relevant from the non-relevant *COVID-19* literature.

## 1 Introduction

The *COVID-19* pandemic has been responsible for almost 10 million people infected worldwide, and has left close to 1 million people dead till mid-September 2020, according to the *World Health Organization*[1]. The whole world observes in awe the catastrophe that the pandemic is leaving behind; human lives, economies and markets have been struck fiercely, as the scientific community and industry, united in all fronts, is seeking for treatment and vaccine solutions against the disease caused by the *2019-ncov* virus.

In these times where scientific advancements are sought and expected rapidly, there lies the challenge of filtering efficiently the scientific literature for the most relevant articles that can help clinicians, nurses, biologists, chemists, bioinformaticians, data scientists and other researchers operating in affiliated disciplines, to combat the pandemic. All these research and practice protagonists have many, and heterogeneous, requirements on what would be a relevant *COVID-19* article; more importantly, they have extremely limited time to scan large volumes of literature.

The risks faced in the aforementioned challenge are multiple; for first, the information, in the form of scientific articles, needs to be **timely**. This requires acceleration of the whole peer-review and publication process for the *COVID-19* relevant articles, to enable the fastest possible communication of breaking scientific and clinical results. The information needs to be **accurate** as well; therefore, without jeopardizing quality, the editors and publishers of scientific content need to have in place fast-track review processes for these important articles. Furthermore, the information needs to be

---

[1]https://covid19.who.int/

**highly relevant** for the aforementioned protagonists who combat the pandemic at the forefront, and have limited time for extensive literature reviews given their crucial duties. Last but not least, the challenge is becoming increasingly difficult, given that the **volume** of the relevant literature for *COVID-19* is continuously growing; indicatively, the *Elsevier* published articles on *COVID-19*[2] have grown in volume from few tens of articles per week in March 2020, to almost $1,000$ articles per week in June 2020. This is an increase of approximately 2 orders of magnitude in a period of 4 months.

In order to avoid "information choking", the community requires efficient data science solutions and respective initiatives that can help researchers navigate through this volume of information and focus on the most relevant articles based on their information need. Some examples of such initiatives, or enablers thereof, are:

- the *TREC-COVID*[3] which follows the well-known to the information retrieval community *TREC* series for building information retrieval test collections, and enabling the development of novel document retrieval algorithms,

- data science challenges organized by *Kaggle*[4], e.g., the *COVID-19 Open Research Dataset Challenge (CORD-19)*[5],

- public releases of *COVID-19* relevant datasets of scientific articles, such as the *CORD-19*[6], or full texts made available by *PMC*[7], in which all scientific publishers contribute, and,

- publicly available and free to use research platforms, where researchers can navigate the *COVID-19*, and all relevant literature, and benefit from advanced text mining and natural language processing solutions, e.g., the *Elsevier's Coronavirus Research Hub*[8].

---

[2]All of the *Elsevier* articles pertaining to *COVID-19* are made available to the community: https://www.elsevier.com/connect/coronavirus-information-center
[3]https://ir.nist.gov/covidSubmit/
[4]https://www.kaggle.com/
[5]https://www.kaggle.com/allen-institute-for-ai/CORD-19-research-challenge
[6]https://allenai.org/data/cord-19
[7]https://www.ncbi.nlm.nih.gov/pmc/about/covid-19/
[8]https://www.elsevier.com/clinical-solutions/coronavirus-research-hub

The majority of the aforementioned initiatives imply the existence of a *COVID-19* scientific article relevancy mechanism, that can filter the core literature on the pandemic, to be included in such collections or data science challenges and platforms. In this paper we present such a framework, namely *CORA*, and we argue that it may constitute the basis for surfacing efficiently the core *COVID-19* literature in a way that it addresses the majority of the information needs of the protagonists who fight the pandemic. The contribution of *CORA* can be summarized in the following: (i) it defines the information need behind relevancy to *COVID-19*, having ingested the feedback of researchers and professionals in medicine, biology, chemistry, bioinformatics and data science, (ii) it offers a benchmark set for the task, with labelled "*relevant*" and "*non-relevant*" *COVID-19* scientific articles, and, (iii) defines an efficient approach that combines search and machine learning, to balance optimally between precision and recall for the task. The impact of such an approach is tremendous; for first, it can help scientific publishers and editors to flag early submitted articles that are core to *COVID-19*, and ensure the acceleration of their review and final publication. It can also be used to filter out large volumes of scientific literature, and retain only the body of the literature that is core to *COVID-19*. This can help accelerate the preparation and production of data science datasets and challenges aiming to address the pandemic. The presented framework is generic, and is described in detail so that it can be reproduced in any environment for these two purposes.

In the remaining of the paper we describe the information need of relevancy to *COVID-19* (Section 2), the process used to create and validate the benchmark set for the training and the tuning of the approach (Section 3), the details of the *CORA* framework (Section 4), as well as results of various methods, including *CORA*, in the produced set (Section 5).

## 2   *COVID-19* Information Need for Relevant Scientific Literature

One of the largest publicly available datasets for *COVID-19*, namely *CORD-19* (Wang et al., 2020), draws its contents from *PubMed Central*, *bioRxiv*, *medRxiv*, and the *World Health Organization* (*WHO*). All the major scientific publishers, such as *Elsevier*, and *Springer Nature* are contribut-

```
"COVID-19" OR "Coronavirus" OR
"Corona virus" OR "2019-nCoV"
OR "SARS-CoV" OR "MERS-CoV"
OR "Severe Acute Respiratory
Syndrome" OR "Middle East
Respiratory Syndrome"
```

Figure 1: The keyword-based query used to retrieve *COVID-19* relevant documents for *CORD-19* from *PubMed Central*, *bioRxiv*, and *medRxiv*. Papers that match on these keywords in their title, abstract, or full text are included in the dataset.

ing to it, and have offered every help for its compilation. In the case of *WHO*, the data can be pulled from a hand-curated database of relevant literature compiled by the organization[9]. However, in the case of the remaining three sources, a generic keyword query is used on the title, abstract and body text of the articles, to filter the ones that are included in the collection. The query is shown in Figure 1.

The query used for the compilation of *CORD-19* includes the fundamental keywords for the pandemic; however, the precision of the aforementioned query is highly arguable. A scientific article may refer to *COVID-19* or any of the coronaviruses for multiple reasons, and often the article can be deemed as irrelevant by expert doctors, biologists and chemists. For example, the article could refer to the financial consequences of *COVID-19*, or to its impact in some social aspects. It could even just refer to *COVID-19* as the most recent example of a pandemic, without discussing about the specific pandemic at all, in a scientific, medical or clinical context. We argue that the expert users who combat the pandemic have an underlying information need that is much more specific than the one expressed from the aforementioned query, and that there should be much more efficient mechanisms to filter the relevant core *COVID-19* articles.

As a first step for the creation of *CORA* we interviewed experts in the field of medicine, biology, chemistry and bioinformatics, who combat the pandemic, and attempted to extract their information needs. This resulted in a number of inclusion and exclusion criteria, that represent the information need, and can be used to compile a benchmark dataset for identifying core *COVID-19* articles. The

---

[9] http://tiny.cc/2n9jrz

inclusion and exclusion criteria are presented in Figure 2.

As it is illustrated, the protagonists who combat the pandemic, are interested exclusively in the diagnosis, treatment, vaccine development, pathology, and virology of *COVID-19*, as well as in literature about other coronaviruses. Furthermore, the experts are also interested in how hospitals are addressing the pandemic, how does the health care systems manage it, and what are some population statistics, and demographics of the disease. All experts were explicit in that, articles related to the impact of the pandemic in areas such as economy, education, transport, etc., are of secondary importance and should not be included in a core scientific *COVID-19* collection, aiming to aid the combat to the disease.

## 3 Preparation of the Benchmark Set

One of the main challenges in any supervised learning task is to have good quality and high volume training data for the algorithms to learn optimally. In cases where training sets are not available, one must create a bespoke data set for the task at hand. Creating a valid, accurate, and large data set is a time-consuming and laborious task. Data sets are typically created by manual annotation of data points, e.g., scientific articles in our case, from a pool of randomly selected data points from a population. The size of the training dataset typically depends on factors such as task complexity, resources, time, and budge availability.

In order to create a benchmark dataset which includes both "relevant" and "non-relevant" articles to the criteria illustrated in Figure 2, as a first step we applied the query illustrated in Figure 3, into the forward flow of the *Elsevier* accepted articles for a period of 2 months. This query can be seen as a much more detailed version of the simple and generic keyword-based query illustrated in Figure 1 that was utilized for the compilation of the *CORD-19* dataset. For the manual annotation of the documents returned by the query, we used active machine learning, and more precisely the general approach described by Konyushkova et al. (Konyushkova et al., 2017). Active learning in this case provides an efficient way of selecting the *right* document sample(s) for labelling. In active learning, an algorithm picks the examples that are more useful in order for the machine learning process to reach its full potential. We used *BioBERT* as the

**Covid-19 Inclusion Criteria**

In principal, all **primary** impact Covid-19 articles will be included, such as (but not limited to):

1. Articles dealing with **diagnosis**, **treatment**, **vaccine development**, **patient social context**, **pathology, virology of** SARS-CoV, MERS-CoV or coronavirus in general
2. Articles about **other corona viruses**, including non-human corona viruses that deal with **vaccine development** or **host-virus interaction**
3. Articles dealing with **hospital-ways to deal with pandemics** or computer models around **pandemic propagation modelling**
4. Articles describing population and **population related phenomena** (includes demography and population statistics) of the Covid-19 pandemic
5. Articles about the impact of Covid-19 on the **health care system** (such as disease management, healthcare facilities and services, healthcare management, healthcare organization, healthcare personnel, and healthcare quality)

**Covid-19 Exclusion Criteria**

In principal, all **secondary** impact Covid-19 articles will be excluded, e.g. articles talking about Covid-19 impact on:

1. economy
2. education
3. transport
4. social media

Figure 2: Inclusion and exclusion criteria of scientific information derived by analyzing the information needs of research experts and practitioners who combat the *COVID-19* pandemic, from the fields of medicine, biology, chemistry, and bioinformatics.

"covid-19" OR "covid 19" OR "covid19" OR "corona virus" OR "coronavirus" OR "corona-virus" OR "corona viruses" OR "coronaviruses" OR "corona-viruses" OR "orthocoronavirinae" OR "coronaviridae" OR "coronavirinae" OR "2019-ncov" OR "2019ncov" OR "2019 ncov" OR "hcov-19" OR "sars-cov" OR "sars cov" OR "severe acute respiratory syndrome" OR "sars-cov-2" OR "sars-cov2" OR "mers-cov" OR "mers cov" OR "middle east respiratory syndrome" OR "middle eastern respiratory syndrome" OR ("angiotensin-converting enzyme 2" AND "virus") OR ("ace2" AND "virus") OR "soluble ace2" OR ("angiotensin converting enzyme2" AND "virus") OR ("ards" AND "virus") OR "acute respiratory distress syndrome" OR ("sars" AND "virus") OR ("mers" AND "virus") OR ("wuhan" AND "virus")

Figure 3: Generic keyword-based query strategy used to compile the corpus for annotation from the Subject Matter Experts in the fields of medicine, biology and chemistry, towards creating a benchmark set for the task.

base classifier in the active learning pipeline.

There are several criteria an algorithm can use to pick the best samples for annotation, such as based on uncertainty, committee, or by bagging and boosting (Olsson, 2009). For this task, we used uncertainty sampling which is one of the popular methods, and is considered to be very efficient (Shen et al., 2017). In uncertainty sampling, the algorithm picks a sample for annotation from the unlabeled pool where it is least confident about its prediction probability. This resulted in a much smaller but more informative data set for our task. The deep active learning algorithm utilized to compile the most useful such set for *CORA* is described in Algorithm 1. We first fine-tuned *BioBERT* on a small expert-curated seed set (~1,600 data points) and measured its accuracy on the test set. Secondly, the algorithm enters the active learning loop, where

the unlabeled data samples are picked using the uncertainty method and human validators are asked to provide labels for the data points. The model is then further fine-tuned on the data points where the classifier's label contradicts human label. The accuracy in line 12 of the algorithm measures how certain the model is on the unlabeled samples. This process continues til the maximum iterations or the active learning algorithm suggest that no further training is required (i.e., desired certainty is achieved). We also created a separate test set to evaluate the performance of the *CORA* machine learning model. Both the training and test set were manually annotated by in-house subject matter experts (Afzal et al., 2020). Table 1 describes the statistics of the benchmark set, for both the training and the test subsets. Figure 5 shows the incremental performance of the classifier on the test set as the

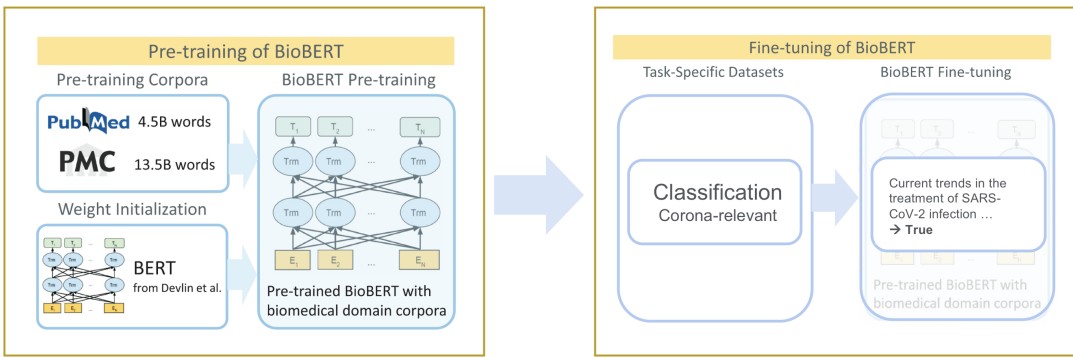

Figure 4: High-level description of the machine learning model used in *CORA* to filter in relevancy the originally retrieved articles from the generic query strategy.

number of labeled samples increases in the training set during active learning process.

## 4 The *CORA* COVID-19 Relevancy Algorithm

*CORA* is aiming at encapsulating the information needs described in Section 2, and retaining an optimal balance between precision and recall in the process of retrieving relevant documents according to these needs. The focus in satisfying the precision to the information needs can be addressed by training a machine learning model on the benchmark set described in Section 3. However, *CORA* needs to start from a much larger set, to also satisfy the requirement that recall is as high as possible; yet such a set needs to minimize the risks of introducing a large number of false positives, and totally irrelevant articles.

In order to achieve this balance, *CORA* utilizes first the keyword-based strategy illustrated in Figure 3, and then applies a machine learning model to filter out the "non-relevant" articles from this originally wide net that was cast to perform the information retrieval. The machine learning model that *CORA* is using is a fine-tuning of *BioBERT* (Bidirectional Encoder Representations from Transformers for Biomedical Text Mining) (Lee et al., 2020) for the task of learning the *"relevant"* and *"non-relevant"* classes from the benchmark set.

*BioBERT* is a *BERT*-based language representation model which is pre-trained on biomedical corpora from *PubMed* and *PMC*, as well as on the English *Wikipedia* and a corpus of books. It has reported state-of-the-art performance in several NLP related tasks on biomedical text, such as named entity recognition, and biomedical question answering (Tsatsaronis et al., 2015). The high level

description of the machine learning model used in *CORA* is illustrated in Figure 4.

Given the benchmark training set obtained via deep active learning, as illustrated in Algorithm 1, the *CORA COVID-19* relevancy can be described in simple steps, and is illustrated in Algorithm 2. The description of the algorithm covers both the preparation and training, as well as the inference steps, given an input set of unseen documents $D_{test}$ to be classified as "relevant" or "non-relevant".

## 5 Experimental Results and Discussion

In this section we present the results of the empirical evaluation on the produced benchmark set described in Table 1. The numbers reported throughout the section refer to the performance of the tested models on the test (unseen) subset of the benchmark document collection. In all cases, *precision*, *recall* and *F1-Score* are reported for both the "relevant" and "non-relevant" classes. Section 5.1 measures performance of two flavors of *CORA*; one that utilizes a *BioBERT* fine-tuning which favors precision, and one that favors recall. We distinguish the evaluation of *CORA* in this set, from other classification algorithms, as the created benchmark set has been produced using deep active learning on the *BioBERT* model, and, therefore, has included examples that are selected to help the fine-tuning of *BioBERT* specifically. Nevertheless, for reasons of completeness, and of scientific clarity, and in order to illustrate the potential value of this set for utilization by other methods, we report in Section 5.2 the performance of several mainstream machine learning models.

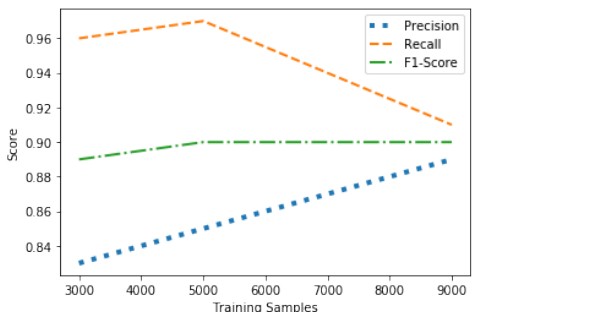 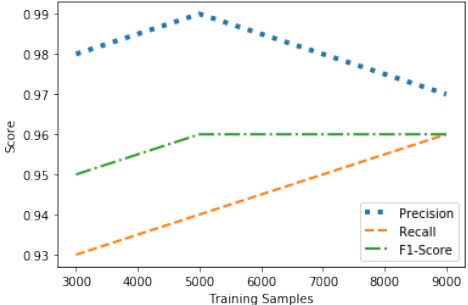

(a) BioBERT learning on relevant class  (b) BioBERT learning on non-relevant class

Figure 5: Biobert fine-tuning learning curves for the "relevant" (Figure 5a) and "non-relevant" (Figure 5b) classes.

## 5.1 CORA Evaluation

The evaluation of *CORA* on the test subset is focusing on measuring the performance of the *BioBERT* fine-tuning. We have fine-tuned two variants of *BioBERT* in *CORA*. The first variant is focusing on maximizing the precision on the positive class ("relevant"), while the second is focused more on the recall. The difference between these two variants can be achieved by doing a grid-search on the threshold for the classification. The results of the evaluation are reported in Table 2. As the numbers in the table suggest, both variants of the *BioBERT* result in a high F1-score, equal or more than 90% for the "relevant" class. The performance for the "non-relevant" class, which is also the majority class in the test set, is even higher, at 96%. The difference between the two variants is small, and, given the volume of the test set, it appears to be statistically insignificant for the precision in the positive class, but significant for the recall.

The high F1-score in both variants and classes, and especially given the imbalance between the two classes in the test set, which is simulating the actual forward flow of articles in the population, designates that the model has learned successfully to distinguish between "relevant" and "non-relevant" *COVID-19* articles. Given also that the inclusion and exclusion criteria for the labeling of the set encapsulate the information needs of the expert users, we can argue that the proposed algorithm manages to filter the relevant *COVID-19* scientific literature in an acceptable manner, with an *F1-score* equal to, or greater than 90%.

As the *CORA* algorithm utilizes both a keyword-based search, and a machine learning model for the final filtering, it is important to highlight the differences in the performance when the machine learning model is not used. This way, we have an indication of the contribution that the model brings into *CORA*, as well as the ability to individually compare different query strategies. For this purpose, in Table 3 we are presenting the performance of the two keyword search strategies discussed in this paper, namely the *CORD-19* keyword query (first row), and the extended keyword query that we have included in *CORA* (second row), on the test set that we have created, with the focus on the "relevant" class. We also present the effect of applying the machine learning model as the final filtering step, on the results of the extended keyword query (third row), in essence reporting on the overall *CORA* performance, discussed in Table 2.

Comparing the first two rows of Table 3 we can see an expected result: the *CORD-19* keyword query provides lower recall than the extended keyword query, with a benefit of a better precision. As the extended keyword query was designed to capture holistically all the information needs illustrated in Figure 3, and, therefore has many more keywords, it returns much larger number of articles, harming the precision, but covering much better the information needs of the experts. The third row shows the great advantage of adding the trained machine learning model for filtering that set; with a loss in recall of 4 p.p., yet still remaining very high at 94%, the model manages to filter out also a lot of false positives coming from the extended query, boosting the precision by 14 p.p., to 88% from 74%. The effect of applying the whole *CORA* algorithm is eventually made fully visible by looking at the differences also in the F1 scores: the addition of the machine learning model contributes 5 additional p.p. to the *CORD-19* keyword query approach, and 7 additional p.p. to the extended key-

**Algorithm 1:** CORAACTIVELEARNING

**Input:** A document collection $D$ of scientific articles; a small labelled training set $L_{t_0} \in D$, with $L = [\text{"relevant"}, \text{"non-relevant"}]$; $t_{max}$ as the maximum number of iterations, $acc$ as the desired accuracy of the model

**Output:** The final training set $L_t$ after $t_{max}$ iterations, or achieved accuracy $acc$

1   $i = 0$
2   $U_{t_i} \leftarrow D \setminus L_{t_i}$
3   train classifier $f_{t_i}$ on $L_{t_i}$
4   measure $acc(f_{t_i})$
5   **while** $i \leq t_{max}$ **and** $acc(f_{t_i}) \leq acc$ **do**
6      pick instance $x_i \in U_{t_i}$ based on *uncertainty sampling*
7      annotate $x_i$ with $L$
8      $L_{t_{i+1}} \leftarrow L_{t_i} \cup x_i$
9      $U_{t_{i+1}} \leftarrow U_{t_i} \setminus x_i$
10      $i \leftarrow i + 1$
11      train classifier $f_{t_i}$ on $L_{t_i}$
12      measure $acc(f_{t_i})$
13   **end**
14   **return** $L_{t_i}$

**Algorithm 2:** CORAALGORITHM

**Input:** A document set $D_{test}$ of unseen scientific articles

**Output:** A list of classification labels $L_{D_{test}}$ from $L = [\text{"relevant"}, \text{"non-relevant"}]$

1   **if** *classifier $f_{L_i}$ not initialized* **then**
2      /* refer to Algorithm 1 */
     $f_{L_i} \leftarrow$ finetune BioBERT on $L_i$
3   **for** $j \leftarrow 1$ **to** $|D_{test}|$ **do**
4      **if** $D_{test}[j]$ *does not satisfy CORA query* **then**
       /* refer to query illustrated in Figure 3 */
5        $L_{D_{test}}[j] \leftarrow$ *"non-relevant"*
6      **else**
7        $L_{D_{test}}[j] \leftarrow L(f_{L_i}(D_{test}[j]))$
8   **end**
9   **return** $L_{D_{test}}$

| | Relevant | Non-Relevant | Total |
|---|---|---|---|
| Training set | 3296 | 5920 | 9216 |
| Test set | 324 | 910 | 1234 |

Table 1: CORA training and test set

word query approach; in the former case the higher F1 contribution is attributed both to increased precision and recall, while in the latter primarily in a very large boost in precision.

## 5.2 Evaluation of other Classification Algorithms

One of the potential drawbacks of a data set generated through active learning is that it's primarily biased towards the preferences of the model used in the loop (i.e., base learner) and peculiarities of the task. It has been questioned whether such a data set can be used effectively by a machine learning algorithm different from the one used as a base learner (Olsson, 2009). Therefore, a direct comparison between *BioBERT* and other classifiers trained on the same set would not be fair, since the training set was generated through an active learning system with *BioBERT* as a base learner.

However, in order to illustrate that the data set captured the underlying characteristics of the data based on our relevancy inclusion and exclusion criteria, we trained and evaluated several other mainstream machine learning classifiers, namely *Support Vector Machines* (*SVM*), *XGBoost*, *Logistic Regression*, and *Naive Bayes*.

The results of these classifiers in the same test set are presented in Table 4. The best performance from this set of classifiers was achieved by *XGBoost*, with a reported precision of $85\%$, recall of $95\%$, and an *F1-score* of $89\%$ in the "Relevant" class. This performance is very close to *CORA*'s *BioBERT*, suggesting that the same set can be very useful for training other classifiers as well, despite the fact that the set was created with a bias to help *BioBERT* resolve the uncertainty between the two classes.

## 6 Conclusions and Future Work

Following the breakout of the *COVID-19* pandemic early in 2020, the scientific community, industry and governments around the world joined forces to combat the spreading of the disease, and to identify efficient treatment methods, as well as vaccine solutions against the *2019-ncov* virus. Efficient and reliable information communication, including the latest scientific advancements in the form of peer-reviewed published articles, has proven to hold

| BioBERT fine-tuned models | | Precision | Recall | F1-Score |
|---|---|---|---|---|
| Precision Favored | Non-Relevant | 0.97 | 0.96 | 0.96 |
| | Relevant | **0.89** | 0.91 | 0.90 |
| Recall Favored | Non-Relevant | 0.98 | 0.95 | 0.96 |
| | Relevant | 0.88 | **0.94** | 0.91 |

Table 2: Performance of two fine-tuned *BioBERT* models on the test set; a precision-favored and a recall-favored version of the model.

| | Class | Precision | Recall | F1-Score |
|---|---|---|---|---|
| CORD-19 Keyword Query | Relevant | 0.85 | 0.86 | 0.86 |
| Extended Keyword Query | Relevant | 0.74 | **0.98** | 0.84 |
| *BioBERT* fine-tuned | Relevant | **0.88** | 0.94 | **0.91** |

Table 3: Performance of *keyword queries* and the fine-tuned *BioBERT* model on the test set.

great challenges; primarily the lack of fast, and accurate ways to focus only on the core *COVID-19* scientific papers and filter out the secondary impact articles.

In this paper we presented *CORA*, an algorithmic solution to filter the relevant scientific papers, and save time from the experts to combat the disease, and focus only on the primary impact information. The contribution of this work is multi-fold: (i) we present a framework of inclusion and exclusion criteria that may be used as guidelines to annotate corpora of scientific publications, towards building benchmark datasets for the purpose of developing and tuning *COVID-19* relevancy systems; the criteria encapsulate the information needs of experts across medicine, biology, chemistry, and bioinformatics, in order to combat efficiently the pandemic, (ii) we applied a simple, yet efficient deep active learning approach to compile such a benchmark set with the help of subject matter experts for the hand curation of the labels; the approach utilized the fine-tuning of *BioBERT* as a base classifier, and we demonstrated that the produced set is also very meaningful for training other classifiers as well, (iii) we introduced the *CORA* algorithmic framework for filtering the relevant scientific literature; *CORA* combines an extensive keyword-based query to initialize a large pool of potentially relevant documents and maximize recall, and a trained fine-tuned *BioBERT* model, to retain only the relevant articles from this pool, (iv) we demonstrated via an experimental evaluation on the benchmark set, that the *CORA* algorithm can achieve 96% *F1-score* on detecting the non-relevant documents, and

91% on detecting the relevant documents, constituting *CORA* a satisfactory solution for production settings of this exercise.

As a future work, we plan to experiment further with novel machine learning models, e.g., *Albert* (Lan et al., 2019), and *Electra* (Clark et al., 2020), who have shown great promise in the *GLUE* leader board results[10], as well as with alternative active learning approaches, e.g., *batch-aware* methods (Chen and Krause, 2013), in order to improve further this performance. More importantly, having the understanding that the terminology around the *COVID-19* literature is evolving fast over time, new terms appear constantly, and the vocabulary is shifting focus towards the names of new promising targets, compounds or characterization of symptoms and treatment options, we will focus in enriching *CORA* with a novel adaption of its keyword-based query over time. By conducting novel keywords extraction from the recent scientific literature, the *CORA* keyword-based query can be enhanced automatically with new terminology. In this manner, the original pool of fetched documents can still satisfy the requirement of very high recall, as they are fetched by a query which follows the vocabulary trends adopted by the published scientific literature on *COVID-19*. Additionally, to help further with the information overload issue, we plan to introduce domain-specific targeted labels for different user groups (e.g., clinicians, bioinformaticians, chemists), allowing any *COVID-19* relevant literature to be potentially filtered according to domain specific information needs.

---

[10] https://gluebenchmark.com/leaderboard

|                     |              | Precision | Recall | F1-Score |
|---------------------|--------------|-----------|--------|----------|
| **SVM-Linear**      | Non-Relevant | 0.90      | 0.74   | 0.81     |
|                     | Relevant     | 0.52      | 0.78   | 0.62     |
| **XGBoost**         | Non-Relevant | **0.98**  | **0.94** | **0.96** |
|                     | Relevant     | **0.85**  | **0.95** | **0.89** |
| **Logistic Regression** | Non-Relevant | 0.92  | 0.80   | 0.86     |
|                     | Relevant     | 0.60      | 0.81   | 0.69     |
| **Naive Bayes**     | Non-Relevant | 0.95      | 0.74   | 0.83     |
|                     | Relevant     | 0.55      | 0.89   | 0.68     |

Table 4: Performance of *Support Vector Machines* (*SVM*), *XGBoost*, *Logistic Regression*, and *Naive Bayes* in the *CORA* test set.

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
