# OpenReview forum: "CORA: A Deep Active Learning Covid-19 Relevancy Algorithm to Identify Core Scientific Articles"
_EMNLP/2020/Workshop/NLP-COVID — NLP-COVID19-EMNLP Oral_

### Official Review · AnonReviewer2 · 2020-09-17
**A well-written paper requires minor clarifications**

**Rating:** 8
**Confidence:** 4

**Review:**

The authors generated a benchmark dataset of relevant scientific information to COVID. They used active learning to build the training set and classified literatures into COVID relevant and non-relevant with high precision and recall.

The paper is well written and includes sufficient technical details. The authors not only evaluated the performance the purposed method, but also evaluated the benchmark dataset using other classifiers.  The classifier fine-tuned on BioBERT outperforms other classifiers.

Please clarify the following information in the paper:

The purpose of using active learning for classification is to reduce the number of labeled samples. Please provide the number of samples labeled by experts in Algorithm 1 and show the performance of classifier with the increase of labeled samples.

The performance of classifier on two relevant and non-relevant sets were reported. In each of set, relevant and non-relevant literatures were included. What is the purpose of generating two sets?

Two datasets were used: training and regression set. Both datasets were annotated by expert. What is the purpose of using two datasets and why only the regression dataset was used to compare with other classifiers?

---

> ### Author Response · Authors · 2020-09-27
> **Re: A well-written paper requires minor clarifications**
>
> We thank the reviewer for reviewing our paper and for providing comments. Please see below for our responses to the points made in the review.
>
> **The purpose of using active learning for classification is to reduce the number of labeled samples. Please provide the number of samples labeled by experts in Algorithm 1 and show the performance of classifier with the increase of labeled samples.**
>
> In total 9216 samples were labeled by the experts with the help of active learning as described in Algorithm 1. This set was used to train our BioBERT classifier. Additionally, 1234 samples were labeled (regression or test set) to evaluate the performance of BioBERT and other classifiers. The training and regression set statistics are described in Table 1. The incremental performance of the classifier (BioBERT) on increasing number of training set is presented in Figure 5. We will add some text in the paper to highlight this. Full data set is available on Mendeley Data and a citation will also be added in the final revision.
>
> **The performance of classifier on two relevant and non-relevant sets were reported. In each of set, relevant and non-relevant literatures were included. What is the purpose of generating two sets?**
>  **Two datasets were used: training and regression set. Both datasets were annotated by expert. What is the purpose of using two datasets and why only the regression dataset was used to compare with other classifiers?**
>
> We created two data sets in this study, one to train the classifiers (referred to as 'Training set' in the paper) and the second to test the performance of the classifiers (referred to as 'Regression set' which is often called a test set). Each of the sets contain COVID relevant and non-relevant samples. To avoid any confusion, we have replaced the term ‘regression set’ with the standard term 'test set'. Similar to BioBERT, other classifiers were also trained on the 'training set' and all of them were evaluated on unseen test set, which is a common practice for unbiased evaluation. We will modify the text to make this point clear as well.

---

### Official Review · AnonReviewer3 · 2020-09-21
**A paper proposing new task and benchmark dataset**

**Rating:** 6
**Confidence:** 3

**Review:**

This paper introduces a task of filtering scientific literature for relevant artciles. (main contribution)
The authors frame the task as a binary classification problem, and use both BioBERT based classifier and conventional classifiers to solve the task.
The second contribution is that the authors release an expert-curated benchamark set for the task.
Active learning is used to develop the training set, where BioBERT based classifier is used as the base classifier.

Strengths
- Although the proposed task focus on meeting the information needs from medicine, biology, chemistry and bioinformatics researchers, the task of filtering out scientific literature should have broader audiences. The proposed framework is generic and should be able to find use cases in other domains.
- The authors mention the dataset is available upon request

Limitations
- The binary classification setting seems oversimplified. The authors have defined several criteria which, in my opinion, can be used as targeted labels, because these criteria seem to target different groups of users. For example, practitioners who focus on treating patients may feel articles beloing to criteria 3 is relevant but those to criteria 5 irrevant. If the task is framed as a multi-class classification problem, it may further solve the overload issue.

Questions and suggestions:
- It would be interesting to see the performance of a simplest baseline that uses only query in Figure 3. In other words, you can use only the IF part of your Algorithm 2 (line 4) and assign all others `relevant'.

- Page 3: 'keyword query is used on the metadata of the articles' vs. Figure 1 caption 'Papers that match on these keywords in their title, abstract, or full text are included in the dataset.'. Are these two conflicting? Is full text used?
- Page 4: Line 4 in Algorithm 1, measuare accuracy on which set?
- Page 4: You didn't mention which classifier you use in Section 3, until in Section 5, you say the BioBERT based classifier is used as the base classifier. Maybe make it clear in Section 3.
- Page 4: is the seperate regression set randomly sampled?
- Page 5: The first part of Algorithm 1 is confusing. Should the training of BioBERT classifier not involve active learning? and the classifier is trained only once on the complete training set? The second part of Algorithm 2 (line 3-8) looks unnecessary.
- Page 6: Table 2, not sure why the bottom line of the first column is missing. Also, you may add \bottomline to Table 3.
- Page 6: not sure what Figure 5 is for, since you didn't mention it at all in the text.
- Table 2, 3: reporting only results on Relevant class should be enough, because it is a binary classification task and relevant class is what we really care.
- Page 7: suggest to simplify Section 6. For example, delete the first paragraph, which seems no need to repeat.
- Page 8: suggest to add citation after batch-aware methods, or explain what it is
- Page 8: 'By conducting novel keywords extraction from the recent scientific literature, the CORA keyword-based query can be enhanced automatically with new terminology.' this part sounds not about future work
- The section 4 may benefit from rewriting. It is a bit of confusing. My understanding is at this stage you fine-tune BioBERT based classifier on the complete training set, and this process does not involve training the classifier several times (such as the while loop in Algorithm 1). Suggest to simplify the description.

---

> ### Author Response · Authors · 2020-09-27
> **Re: A paper proposing new task and benchmark dataset - ANSWERS PART 2**
>
>
> -- continued from part 1 --
>
> **Page 8: suggest to add citation after batch-aware methods, or explain what it is**
>
> Thank you for the feedback. We will add citation for batch-aware method.
>
> **Page 8: 'By conducting novel keywords extraction from the recent scientific literature, the CORA keyword-based query can be enhanced automatically with new terminology.' this part sounds not about future work**
>
> This statements builds on the future work statements prior to this line “…we will focus in enriching CORA with a novel adaption of its keyword-based query over time”. The statement in question only refers to the value it may bring.
>
> **The section 4 may benefit from rewriting. It is a bit of confusing. My understanding is at this stage you fine-tune BioBERT based classifier on the complete training set, and this process does not involve training the classifier several times (such as the while loop in Algorithm 1). Suggest to simplify the description.**
>
> Thank you for the feedback. We will simplify the description to make the process and the algorithm clear. In short, first we finetune BioBERT classifier on the seed data, before the active learning process.
> Secondly, in the active learning loop, the unlabeled data samples are picked using the uncertainty sampling method and the human validator is asked to provide the labels for the data points. The model is then updated (finetuned) on the data points where the classifier’s label contradicts human label. This process continues till active learning algorithms suggest that no more training is required (i.e. desired certainty is achieved). We will modify the text and/or algorithm to make it clear in the final revision.

---

> ### Author Response · Authors · 2020-09-27
> **Re: A paper proposing new task and benchmark dataset - ANSWERS PART 1**
>
>
> We thank the reviewer for reviewing our paper and for providing comments. Please see below for our responses to the points made in the review.
>
> **Limitations**
>
> **The binary classification setting seems oversimplified. The authors have defined several criteria which, in my opinion, can be used as targeted labels, because these criteria seem to target different groups of users. For example, practitioners who focus on treating patients may feel articles belonging to criteria 3 is relevant but those to criteria 5 irrelevant. If the task is framed as a multi-class classification problem, it may further solve the overload issue.**
>
> A primary impact Covid-19 article may include more than one of the inclusion criteria of Figure 2. Additionally, analyzing the information need of several important stakeholders combating the pandemic, it became clear that they often need to look at the information in slightly broader yet relevant perspective. Therefore, for this study, we defined the task as a binary classification task to identify primary impact articles containing information needs of one or more groups of users altogether. Having said that, we do agree with reviewer’s feedback that having more find-grained labels (framing as multi-class multi-label task) would further help the information overload issue.  We have already planned this work and would mention in the future work section.
>
> **Questions and suggestions:**
>
> **It would be interesting to see the performance of a simplest baseline that uses only query in Figure 3. In other words, you can use only the IF part of your Algorithm 2 (line 4) and assign all others `relevant'**
>
> We do already have these results and we will put the results in the table where we compare all the models
>
> **Page 3: 'keyword query is used on the metadata of the articles' vs. Figure 1 caption 'Papers that match on these keywords in their title, abstract, or full text are included in the data set.'. Are these two conflicting? Is full text used?**
>
> For the **CORD-19 data set**, all papers matching Figure 1 keywords in their title, abstract, and body text are included as mentioned in the Figure 1 caption.  We will rectify the mistake in the text. Thank you for pointing out.
>
> **Page 4: Line 4 in Algorithm 1, measure accuracy on which set?**
>
> In line 4, we used regression test set to measure accuracy of the classifier initially trained on the seed set (line 3). This is done to identify classifier baseline and to check how much improvement do we get after putting Active Learning in the pipeline. In line 12 accuracy measure means how certain the model is on the unlabeled samples. We will add some text in the manuscript to make the algorithm clear.
>
> **Page 4: You didn't mention which classifier you use in Section 3, until in Section 5, you say the BioBERT based classifier is used as the base classifier. Maybe make it clear in Section 3.**
>
> Thank you for your feedback. We will do the same.
>
> **Page 4: is the seperate regression set randomly sampled?**
>
> Yes, the regression set (test set) was randomly sampled.
>
> **Page 5: The first part of Algorithm 1 is confusing. Should the training of BioBERT classifier not involve active learning? and the classifier is trained only once on the complete training set? The second part of Algorithm 2 (line 3-8) looks unnecessary.**
>
> We first train the BioBERT model on seed data (line 3). This seed data has ~1600 data points. This seed data was manually created by the domain experts. The same BioBERT model, initially finetuned on the seed set, is then used in the active learning. Line 1-4 showcases the BioBERT finetuned on seed data and from line 5 to 12 active learning part is shown. We will add some text in the manuscript to make the algorithm clear.
>
> **Page 6: Table 2, not sure why the bottom line of the first column is missing. Also, you may add \bottomline to Table 3.**
>
> Thank you for the feedback. We will do the same
>
> **Page 6: not sure what Figure 5 is for, since you didn't mention it at all in the text.**
>
> Figure 5 shows the precision, recall and f1 scores on regression test after the model was ingested with 3k, 6k and 9k data points. Figure 5a shows the results for relevant class and 5b shows the results for non-relevant class. This showcase the progress of the model with the help of active learning. We will add this text in the manuscript as well with reference to Figure 5.
>
> **Table 2, 3: reporting only results on Relevant class should be enough, because it is a binary classification task and relevant class is what we really care**
>
> We also added non-relevant results for the sake of completeness and to help readers understand the system fully.
>
> **Page 7: suggest to simplify Section 6. For example, delete the first paragraph, which seems no need to repeat.**
>
> Thank you for the feedback. We will do the same.

---

### Official Review · AnonReviewer4 · 2020-09-29
**Well-motivated dataset work.**

**Rating:** 6
**Confidence:** 4

**Review:**

This paper proposes a dataset that is annotated via a standard active learning framework, aiming at providing supervision for models to filter COVID-19 articles that are out of expert interests. This paper is well-motivated and well-written. Some elaborations appear to be unnecessarily extensive.

1. When the paper says "...however, the precision of the aforementioned query is highly arguable", there should be some statistics to justify this point. Afterall, this point establishes the motivation of this paper. Otherwise, the contribution is limited.

2. In Algo1, line 5, "i<= max" should be "i <= t_max". Further, it would be useful to present a curve of the acc(f_i) over time. This would illustrate the effectiveness of using active learning. It would be interesting to see if this is a log-like curve or some bumpy curve.

3. In footnote 10, it seems the data is ONLY available during the review time? I want to make sure if this is the case. For a closed dataset, even if it is available during the reviewing period, I would give it an immediate rejection, no matter how good the work is.

4. What is Fig 5? Not mentioned anywhere.

5. It would be perfect if this paper can even gently touch on this point: What is the impact of this dataset on other COVID challenges? This would highlight the usefulness of this data. There could be some domain issue that hinders its application elsewhere. But who knows.

-- PS --
While my rating is leaning positive to acceptance, it depends on the authors' response on point 3 above.

---

> ### Author Response · Authors · 2020-09-29
> **Re: Well-motivated dataset work**
>
> Thank you for the review and the feedback. Please see below for our responses to the points made in the review.
>
> **When the paper says "...however, the precision of the aforementioned query is highly arguable", there should be some statistics to justify this point. Afterall, this point establishes the motivation of this paper. Otherwise, the contribution is limited.**
>
> We will surely add numbers in the revised manuscript to support this statement.
>
> **In Algo1, line 5, "i<= max" should be "i <= t_max". Further, it would be useful to present a curve of the acc(f_i) over time. This would illustrate the effectiveness of using active learning. It would be interesting to see if this is a log-like curve or some bumpy curve.**
>
> Thank you for pointing out the typo, we will correct this. We have shown the plots in Figure 5 to showcase the progress of the model with the help of active learning.  We presented precision, recall, and f-score on regression test set after the model was ingested with 3k, 6k, and 9k data points. We will add this missing text in the manuscript as well with reference to Figure 5.
>
> **In footnote 10, it seems the data is ONLY available during the review time? I want to make sure if this is the case. For a closed dataset, even if it is available during the reviewing period, I would give it an immediate rejection, no matter how good the work is.**
>
> We have already made our dataset available publicly on Mendeley Data (https://data.mendeley.com/datasets/yv46ndw67j/1). We will replace the footnote in the paper with a proper reference to the data set.
>
> **What is Fig 5? Not mentioned anywhere.**
>
> Figure 5 shows the precision, recall and f1 scores on regression test after the model was ingested with 3k, 6k and 9k data points. Figure 5a shows the results for relevant class and 5b shows the results for non-relevant class. This showcase the progress of the model with the help of active learning. We will add this text in the manuscript as well with reference to Figure 5.
>
> **It would be perfect if this paper can even gently touch on this point: What is the impact of this dataset on other COVID challenges? This would highlight the usefulness of this data. There could be some domain issue that hinders its application elsewhere. But who knows.**
>
> One of the main datasets (i.e. CORD-19) used in other challenges is primarily a result of the keyword-based query we have shown in Figure 1. As an outcome, the CORD-19 data set also contain non-relevant and secondary impact Covid-19 papers for experts combating the pandemic.
>
> We see at least two straight possibilities where our work can be used in wider Covid-19 related initiatives.
>
> 1)	The strength of our data set is that captures the underlying characteristics of the data based on our relevancy inclusion and exclusion criteria as shown in Figure 2. As it can be seen in Table 3 that even the standard machine learning algorithms (e.g. XGBoost) are able to learn the boundary between relevant and non-relevant Covid-19 articles with high level of precision and recall. Any model train on our data set can be used to first label a more generalized Covid-19 data set (such as CORD-19) and use labels as features in any downstream tasks.
> 2)	The data set itself can be combined with existing Covid-19 (challenge) data sets and the downstream tasks can use the labeled information (relevant and non-relevant) as an *additional feature* to further optimize their solutions.
>
>
> We did not get around to try the above possibilities, but these are somethings we are interested in as well and will definitely be looking forward to do in our future work. We will add the same in the revised manuscript.